# Genome-wide association study of early liveweight traits in fat-tailed Akkaraman lambs

**Mehmet Ulas Cinar**[1,2]*, **Korhan Arslan**[3], **Md Mahmodul Hasan Sohel**[3,4], **Davut Bayram**[5], **Lindsay M. W. Piel**[6], **Stephen N. White**[2¤], **Fadime Daldaban**[3], **Esma Gamze Aksel**[3], **Bilal Akyüz**[3]

**1** Faculty of Agriculture, Department of Animal Science, Erciyes University, Kayseri, Turkiye, **2** Department of Veterinary Microbiology & Pathology, College of Veterinary Medicine, Washington State University, Pullman, Washington, United States of America, **3** Faculty of Veterinary Medicine, Department of Genetics, Erciyes University, Kayseri, Turkiye, **4** Department of Life Sciences, School of Environment and Life Sciences, Independent University, Dhaka, Bangladesh, **5** Faculty of Veterinary Medicine, Department of Animal Science, Erciyes University, Kayseri, Turkiye, **6** USDA-ARS Animal Disease Res. 3003 ADBF, WSU Pullman, Pullman, Washington, United States of America

¤ Current address: Genus plc, DeForest, Wisconsin, United States of America
* mucinar@erciyes.edu.tr

**Data Availability Statement:** All relevant data are within the paper and its Supporting Information files.

## Abstract

Small ruminants, especially sheep, are essential for sustainable agricultural production systems, future food/nutrition security, and poverty reduction in developing countries. Within developed countries, the ability of sheep to survive on low-quality forage intake could act as buffer against climate change. Besides sheep's importance in sustainable agricultural production, there has been less ongoing work in terms of sheep genetics in Near East, Middle East and in Africa. For lamb meat production, body weight and average daily gain (ADG) until weaning are critical economic traits that affects the profitability of the industry. The current study aims to identify single nucleotide polymorphisms (SNPs) that are significantly associated with pre-weaning growth traits in fat tail Akkaraman lambs using a genome-wide association study (GWAS). A total of 196 Akkaraman lambs were selected for analysis. After quality control, a total of 31,936 SNPs and 146 lambs were used for subsequent analyses. PLINK 1.9 beta software was used for the analyses. Based on Bonferroni-adjusted *p*-values, one SNP (*rs427117280*) on chromosome 2 (OAR2) had significant associations with weaning weight at day 90 and ADG from day 0 to day 90, which jointly explains a 0.8% and 0.9% of total genetic variation respectively. The *Ovis aries* natriuretic peptide C (*NPPC*) could be considered as a candidate gene for the defined significant associations. The results of the current study will help to increase understanding of the variation in weaning weight and ADG until weaning of Akkaraman lambs and help enhance selection for lambs with improved weaning weight and ADG. However, further investigations are required for the identification of causal variants within the identified genomic regions.

**Funding:** BA. This research was funded by Erciyes University Scientific Research Projects Coordination Unit under grant number TOA-2018-8282. https://bapsis.erciyes.edu.tr/Default2.aspx. The funder had no role in study design, data collection and analysis, decision to publish, or preparation of the manuscript.

**Competing interests:** The authors have declared that no competing interests exist.

## Introduction

The wild ancestors of sheep have first domesticated approximately 11,000 years ago in the present-day countries Turkey, Iran, and Iraq [1]. Sheep are bred by human for four main products: meat, milk, skins, and wool/hair [2]. The demand for lamb meat is increasing and meat production in the sheep industry is getting more popular day by day, therefore, traits like growth and meat production have received the attention of breeders and geneticists. Although there has been a global decline in sheep numbers in the last two decades, due to a decline in pastoralism and lower feed conversion rates as compared to poultry and pigs, sheep remain a major product in areas which have difficult to manage climates [3]. This is especially true in the semi-arid areas and highlands of Turkey [4]. Breeding lambs in Turkey is very important due to two reasons: first, the consumer's preference for mutton over beef or poultry meat, and second, sacrificing them in religious ceremonies. Fat-tailed Akkaraman sheep are one of the most important breeds in Turkey. Although this breed produces relatively high amounts of mutton and wool, it has generally low productivity in terms of growth traits as compared to imported breeds. However, this breed exhibits a higher resistance to local diseases and has a moderate daily weight gain during fattening (206 ± 0.4 g/day) [5].

Body weight is one of the most important growth and development components in sheep as it both directly and indirectly influences lamb meat yield, wool production, and reproduction. Support of the importance of weaning weight for reproductivity comes from the high (0.80) genetic correlation between these two components [6]. However, breeders not only look at the number of lambs born alive, but also consider the number of lambs weaned and weaning weight when selecting replacement animals [6,7]. Thus, an increase in the growth rate of lambs from birth to weaning has the highest relative economic importance among growth traits [8]. This situation encourages us to selectively breed sheep for liveweight at weaning.

Estimated heritability for lamb daily weight gain and weaning weight was low to moderate depending on the sheep breed studied [9,10]. Compared to other livestock species, the overall number of quantitative trait loci (QTL) and quantitative trait nucleotides (QTN) detected for sheep has been relatively low [11]. Nevertheless, different QTL regions were detected for weaning weight or daily weight gain in various sheep breeds across the world [12–15].

For understanding the genetic background of economical traits, QTL are one of the most useful tools in livestock genetics. In the last couple of decades, several QTLs have been found by genome scanning technology and candidate gene approaches in livestock. Numerous QTL studies for different quantitative traits have been performed on pigs, cattle, and chickens [16]. It is important to note that very few QTLs (4,072) have been identified using a genome scan based on marker-QTL linkage analysis in sheep while the number in cattle and pigs is 177,199 and 34,342, respectively [16]. Furthermore, when the identified QTL confidence intervals in sheep were relatively long due to genotyping technologies such as microsatellites which made difficult to identify the specific genes; nowadays high throughput SNP genotyping technologies provides better insight for detection of genes and variants associated with target quantitative traits [12].

Genome-wide association studies (GWAS) have widely been applied to identify and localize genes related to quantitative traits in several economically important breeds compared to indigenous breeds in farm animals [12]. Additionally, the use of genomic tools for animal breeding in developing countries are less than developed countries [17]. Lack of cheap phenotyping tools and weak data management, adverse environmental conditions (e.g. housing, pasture, climate) for fitness and relatively less experience in advance quantitative genetics may limit the use of genomic tools in breeding of local breeds [17]. The application of high throughput SNP genotyping technologies in association studies would bring more ideas to

increase the efficiency of animal breeding and selection [16]. However, there are relatively few GWAS reports on growth related traits in sheep and no report on Akkaraman sheep. Therefore, the current study aims to conduct a GWAS to identify significant SNPs that are associated with the body weights and body weight gains until weaning, using the data from 146 Akkaraman sheep genotyped with the Illumina Ovine SNP50 BeadChip.

## Materials and methods

### Animals and phenotypic data

A total of 192 Akkaraman male lambs of four nucleus herds were used to record body weights (BWs) until weaning in the Kayseri province of Turkey. All experimental procedures were performed in accordance with the guidelines of the Local Ethics Committee for Animal Experiments at Erciyes University (#13/130-13.11.2013). No animal was sacrificed or subjected any kind of suffering. Commercial husbandry practices were applied during experiment. All the ewes were two years of age. The flocks were fed on pasture except for the months of February and March, which coincides with the highest rate of lambing. During housing, animals were fed a ration composed of wheat stubble, barley, and alfalfa. The mating period extends from the second half of August to late September, where ewes are randomly assigned to rams in field conditions. Lambing started in late December and continued until March. The date of birth of all lambs was recorded and immediately after lambing, they were ear-tagged. From 15 days forth, creep feed (*ad libitum*) was supplied to lambs, and they were weaned at an approximate age of 90 ± 2 days. The lambs were weighed at birth (T0), and on the 30th (T30), 60th (T60) and 90th (T90) day as weaning weight. Average daily gains (ADG) were calculated taking the weight a lamb has gained since the last weight and dividing the weight by the number of days since that last weight. For instance, ADG0_90 was the ADG between birth weight and T90. Each lamb was weighed at least twice by the same person to reduce manmade errors in BW measurements and the average value was considered as the final measurement. BWs were taken in a quiet, relaxed, state in the four herds of Akkaraman sheep. Approximately 10 ml of blood was taken from the jugular vein using vacutainer tubes containing EDTA as an anticoagulant. Genomic DNA was isolated from blood samples using a commercial kit according to the manufacturer's instructions (Qiagen, Hilden, Germany). NanoDrop ND-1000 spectrophotometer (NanoDrop Technologies, Wilmington, DE, USA) was used to check the quantity and quality of the isolated DNA. DNA samples were brought to a final concentration of 50 ng/μl (as required by the Illumina Infinium protocol) and stored at 4°C until use.

### Genotyping and quality control (QC)

Ovine SNP50 BeadChip (Illumina Inc., San Diego, CA, USA) was used to genotype individual SNPs that include 54,241 SNPs. After genotyping all SNP markers were mapped to *Ovis aries* genome build Oar_rambouillet_v1.0 [18]. After that data entry was made into Genome Studio (v.1.9.4) software for further analysis. PLINK v.1.9 software package was used to analyze genotyping data [19]. Quality control of the genotype data was conducted at both the subject level and the SNP level. The individuals or SNPs were removed before final analysis if they met one or all the following criteria: (1) SNP call rate < 0.95, (2) no chromosomal or physical location, (3) individual call rate < 0.95 and/or (4) minor allele frequency (MAF) < 0.05. Following the removal of the SNPs with a linkage disequilibrium (LD) pruning step, all SNPs that are located on sex chromosomes and mitochondrial DNA were discarded in the QC process. SNPs were excluded if a *p*-value of Fisher's exact test for Hardy–Weinberg equilibrium is less than 0.001. After performing a careful QC process, a total of 146 subjects and 31,936 SNPs were included in the analysis data set. Afterwards, the genotypes from the software output were sorted,

proofread, analyzed, corrected, and/or eliminated to obtain the characteristics of the genome-wide data. A genome-wide genotyping system was used to type nucleic acids and Genome Studio software was used to visualize the genotyped data.

## Population stratification analysis

Genome-wide estimates of identity-by-descent (pi-hat) were used to select distantly related individuals. Genotyped lambs were randomly ordered and selected if they were related to every other member of the group below a set threshold. A pi-hat threshold of ≤0.26 was used for lambs included in the association analyses ("sibships of size one"), as this removes duplicate or closely related pairs of lambs. This selection process was repeated for 1,000 iterations, after which the largest set of lambs was identified for further use. In order to assess the population structure, Two-dimension reduction routines provided by PLINK 1.9 were performed: first, variance-standardized relationship matrix was used generate the "pca" for principal components analysis (PCA) [20,21], and the raw Hamming distances was used to generate "mds-plot" for multidimensional scaling (MDS) (S1 Fig) [22]. In order to help correcting population stratification, covariates in association analysis regressions were employed considering top principal components. On the other hand, genetic distance was visualized using MDS coordinates. Clustering of samples were identified by plotting component 1 values against component 2 values using standard classical (metric) multidimensional scaling.

## Genome-wide association study

Since the pedigree of most lambs in the population under study was unknown, GWAS was performed in PLINK v1.9 using linear regression models with liveweights and ADGs between different time points as the response variables and the SNPs as explanatory variables. Association tests were performed based on the least-square (LS) regression tests, where the first 17 principal components were used as covariates for unrelated lambs, farms (where samples are collected), and birth types. Significantly associated SNPs ($p<0.05$) were identified at both the genome-wide and chromosome-wide levels following Bonferroni testing correction. The cut-off for significance was calculated using 0.05/n or 1/n, where n (n = 31,936) is the number of SNPs analyzed at the genome-wide or chromosome-wide level. Thus, the 5% threshold for chromosome-wide significance was $1.565 \times 10^{-6}$. Quantile-quantile plots (Q-Q plot) were visualized by plotting the distribution of obtained vs. expected genome-wide $p$-values (S2 Fig). The "qqman" package in R version 3.5.1 (R core team, 2018) was used to draw Manhattan Plot and Q-Q plot. In order to identify the genome-wide significant SNPs, the Ensembl database using the gene annotation information of *Ovis aries* genome version Oar_rambouillet_v1.0 [18]. The genomic locations were downloaded from https://www.animalgenome.org/sheep/ [16] and genes within the identified region are considered candidate genes.

## Genotyping of *rs427117280* SNP by TaqMan probe

Genotyping of *rs427117280* SNP (OAR chromosome 2: 248863817G>T) using TaqMan SNP Genotyping Custom Assays (Invitrogen, Carlsbad, CA, USA) was performed in 373 Akkaraman lambs. For genotyped lambs, husbandry conditions and number of farms were same as mentioned under the subtitle of animals and phenotypic data. Briefly, genotyping was performed using allelic discrimination primers (Invitrogen, Carlsbad, CA, USA) and hydrolysis probes (Invitrogen, Carlsbad, CA, USA) for *rs427117280* (forward: `ACGTTCTTTAAGCAC ACCCGTTTA`, reverse: `GTTGTAGTCCGCATTGAGAAACG`, `VIC-ACCTTGCTGCACCGCA`, `FAM-AACCTTGCTTCACCGCA`). The remainder of the reaction consisted of TaqMan Genotyping PCR Master Mix (Applied Biosystems, Foster City, CA, USA) and 50 ng of DNA, where

the final reaction volume was 10 μl. The reaction was run in an ABI StepOnePlus Real-Time PCR System (Applied Biosystems, Foster City, CA, USA). Genotyping assays aimed for 95% genotyping rates in these data sets. An online software (http://www.husdyr.kvl.dk/htm/kc/popgen/genetik/applets/kitest.htm) was used to analyze the Hardy-Weinberg equilibrium (HWE) and allele frequency for each SNP and statistical significance was defined as $p<0.05$. The genotype-phenotype association was analyzed with a generalized linear model using the GLM procedure in SAS v9.0. The reduced model included fixed effects of farm (four levels), birth type (single or twin), and genotype. Genotypic comparisons were reported following Tukey-Kramer adjustment, and a $p\leq0.1$ was considered significant.

## Results

The descriptive statistic values of body weights and body weight gains until weaning are presented in S1 Table. After genomic QC, we excluded the following: (1) 3,781 SNPs with call rate <95%; (2) 20 SNPs not in accordance with the Hardy–Weinberg equilibrium; (3) 4,238 SNPs with a minor allele frequency <0.05; (4) and finally a total of 14,266 SNPs were removed following LD pruning, removal of the SNPs on sex chromosomes and mtDNA and SNPs that did not map to any chromosome. Additionally, 46 lambs were excluded due to low heterozygosity or high relatedness among the lambs. After quality control, 146 lambs and 31,936 independent SNPs were used for the final analysis. Based on the remaining number of independent SNPs, $1.565 \times 10^{-6}$ (0.05/31,936) and $3.13 \times 10^{-5}$ (1/31,936) were regarded as genome-wide and chromosome-wide suggestive significant values, respectively.

In total, 5 SNPs reached the suggestive significance level (Table 1), where assessment of the associations showed that most investigative traits were represented. Manhattan plots were used to visualize the five resultant associative traits (Fig 1). As seen in Table 1, two significant loci were identified on chromosome 2 (OAR2) in which one genome-wide and one chromosome-wide significant SNP was detected (Fig 1). Using the sheep genome version *Ovis aries* Oar_rambouillet_v1.0 as a reference on the Ensembl Release 105 (accessed 13 January 2022) was used to query within a distance of 500 Kb for genes surrounding these suggestive markers.

**Table 1. Significant SNPs associated with phenotypic values for body weight and body weight gains until weaning in Akkaraman lambs, and close genes.**

| SNP[1] | Trait[2] | OAR[3] | Position (bp)[4] | $p$-value [5] | Gene[6] | Distance (bp) | Candidate gene[7] |
|---|---|---|---|---|---|---|---|
| _rs427117280_ | T90 | 2 | 248863817G>T | $2.70 \times 10^{-9}$ | ENSOARG00020011332 | Within | *Ovis aries* natriuretic peptide C (*NPPC*) |
| _rs427117280_ | ADG0_90 | 2 | 248863817G>T | $4.04 \times 10^{-12}$ | ENSOARG00020011332 | Within | *Ovis aries* natriuretic peptide C (*NPPC*) |
| rs406101124 | Birth Weight | 7 | 63200764G>A | $2.25 \times 10^{-5}$ | Centrosomal Protein 152 (*CEP152*) | Within | Fibrillin 1 (*FBN1*) |
| rs428733788 | T30 | 1 | 137414362T>C | $6.66 \times 10^{-6}$ | | Intergenic variant | ADAM metallopeptidase with thrombospondin type 1 motif 1 (*ADAMTS1*) |
| rs430444267 | ADG0_30 | 2 | 71582144A>C | $2.23 \times 10^{-5}$ | MAM Domain Containing 2 (*MAMDC2*) | Within | |

[1] Novel QTL (SNPs) are underlined.

[2] T90, liveweight 90 days after birth; ADG0_90, average daily gain from birth to day 90; T30, liveweight 30 days after birth; ADG0_30, average daily gain from birth to 30 days of age.

[3] Chromosomal and

[4] base pair positions of the associated SNPs on *Ovis aries* Oar_rambouillet_v1.0 assembly.

[5] $p$-values calculated from the mixed linear model analysis.

[6] Gene names starting with ENSOARG represent Ensembl nomenclature whereas other gene symbols indicate HUGO nomenclatures.

[7] Candidate genes within 500 kb upstream and downstream of the significant SNPs.

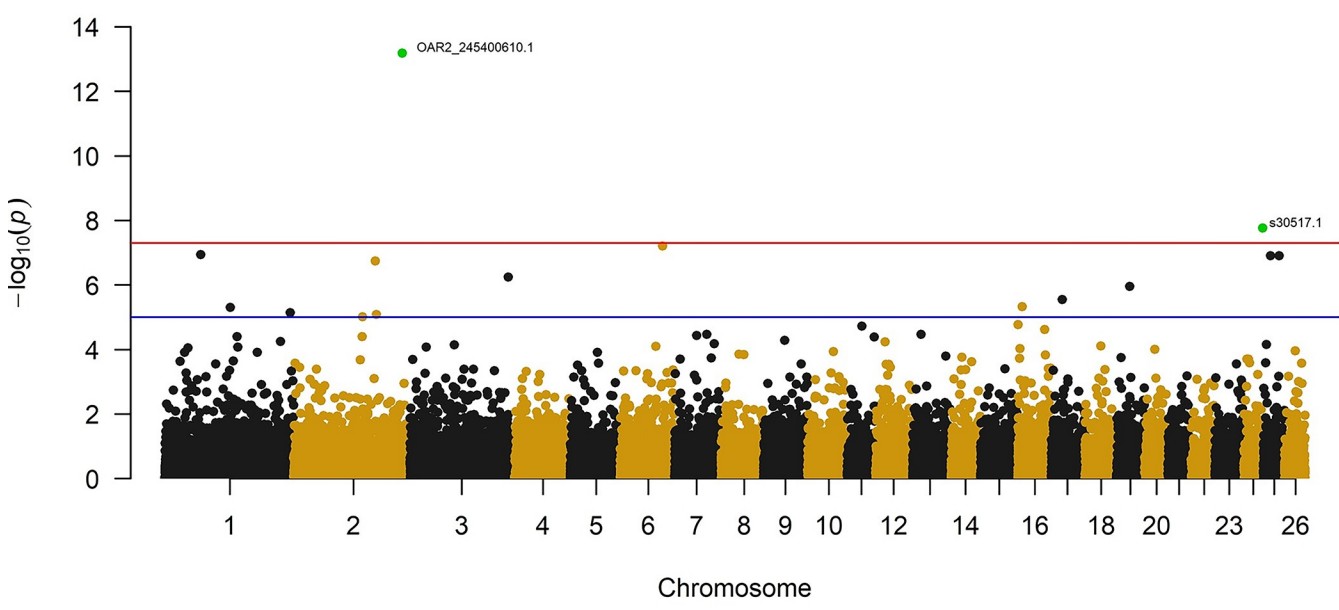

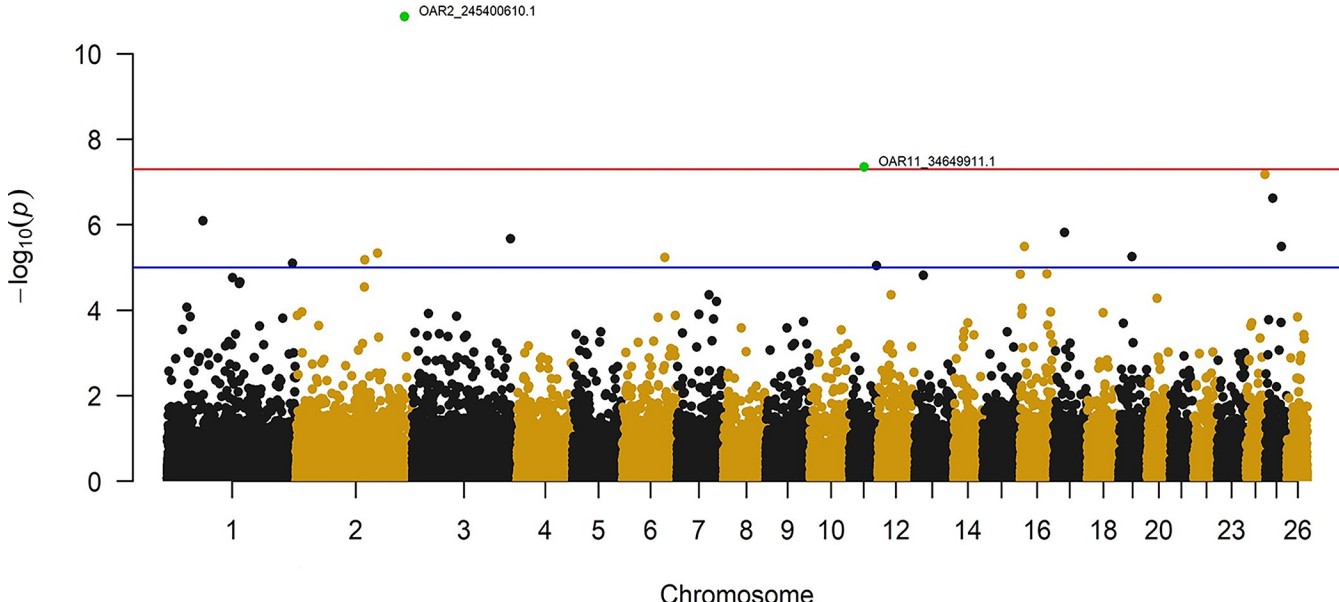

**Fig 1. Manhattan plots of the GWAS for T90 and ADG0_90 traits in Akkaraman lambs.** In the Manhattan plots, Bonferroni adjusted −log10 $p$-values of the quantified SNPs were plotted against their genomic positions; different colours indicate SNPs on different chromosomes from chromosome 1 to 26. T90 = liveweight at 90[th] day after birth and ADG0_90 = average daily gain between birth and 90[th] day.

This led to the identification of candidate gene *NPPC* (*Ovis aries* natriuretic peptide C) in association with the highest significance SNP, where this SNP was found to be significantly associated to both the T90 and ADG0_90 traits. The second SNP on OAR2, *rs430444267*, was

**Table 2. Genotype and allele frequencies at the *rs427117280* SNP.**

| SNPs | Genotypes (Frequencies) | | | Allele (Frequencies) | | HWE (χ2) |
|---|---|---|---|---|---|---|
| *rs427117280* | TT | TG | GG | T | G | 1.414[NS] |
|  | 338 (0.911) | 33 (0.088) | 2 (0.001) | 0.951 | 0.049 |  |

[NS]: Non-significant.

identified as being significant at a chromosome-wide level (Table 1). This SNP is located within the gene *MAMDC2* (MAM Domain Containing 2) and associated with the ADG0_30 trait. While this SNP was located within *MAMDC2*, there is no evidence to suggest that this gene is responsible for growth traits (Table 1). Further searching for genes within 500 Kb did not isolate any candidate genes. Beyond OAR2, SNP *rs406101124*, located within an intron of *CEP152* (Centrosomal Protein 152) on OAR7, which was determined to have chromosome-wide significance associated with the birth weight trait (Table 1). Reference genome query resulted in identification of candidate gene *FBN1* (Fibrillin1). The last suggestive significant SNP for T30 trait (Table 1) was located within an intergenic region on OAR1. The candidate gene *ADAMTS1* (Metallopeptidase with Thrombospondin Type 1 Motif 1) was located close to this region.

GWAS showed that the *rs427117280* SNP was significantly associated with traits T90 and ADG0_90 in Akkaraman lambs. Therefore, SNP *rs427117280* was subjected to genotyping, population genetic analysis, and further association analysis within a larger Akkaraman sheep population. Three genotypes for SNP *rs427117280* were found. TT was the most frequent genotype with a frequency of 0.911. Consequently, the TG (0.088) and GG (0.001) frequencies were much lower. The most frequent allele was T, with an allele frequency of 0.951, which was higher than that of G at 0.049. The $\chi^2$ value was 1.414, confirming that the frequency distribution of SNP *rs427117280* was in accordance with the Hardy–Weinberg equilibrium (HWE) law in the Akkaraman sheep population (Table 2).

Analyzing SNP *rs427117280* at the genotype level identified four significant associations between the SNP and T90, ADG0_90, ADG30_60, ADG60_90 and ADG30_90 traits (Table 3). Specifically, GG genotype lambs were found to be heavier, and their weight gain was higher, as compared to the other two genotypes (Table 3). Notably, statistical analysis exhibited a significant association between genotypes and measured traits in the larger Akkaraman sheep population ($p<0.1$).

## Discussion

Most of the important economic traits in livestock maintain a polygenic complex genetic architecture. Therefore, research focusing on the responsible candidate genes underlying these

**Table 3. Association of *ENSOARG00020011332 rs427117280* with growth traits weights showing adjusted means and standard errors from Akkaraman population.**

| SNP | Genotypes | Birth Weight | T30 | T60 | T90 | ADG0_30 | ADG0_60 | ADG0_90 | ADG30_60 | ADG60_90 | ADG30_90 |
|---|---|---|---|---|---|---|---|---|---|---|---|
| *rs427117280* | GG | 4.6 | 11.0 | 20.52 | 32.05 | 0.36 | 0.34 | 0.35 | 0.31 | 0.38 | 0.35 |
|  | TG | 4.17 | 10.60 | 18.08 | 27.69 | 0.35 | 0.30 | 0.30 | 0.25 | 0.32 | 0.28 |
|  | TT | 4.36 | 10.29 | 17.72 | 26.84 | 0.34 | 0.29 | 0.29 | 0.25 | 0.30 | 0.27 |
|  | Birth type | *** | *** | *** | *** | *** | *** | *** | *** | ** | ** |
|  | Farm | *** | *** | *** | *** | *** | *** | *** | *** | *** | *** |
|  | $R^2$ | 0.42 | 0.37 | 0.38 | 0.37 | 0.37 | 0.38 | 0.37 | 0.30 | 0.24 | 0.32 |
|  | *p*-value | 0.21 | 0.46 | 0.23 | **0.06** | 0.46 | 0.23 | **0.06** | 0.17 | **0.07** | **0.05** |

Bold lettering highlights when genotypes were significantly associated with traits of interest ($p<0.1$).

traits is always intriguing and crucial to support livestock genetics and breeding [23,24]. To the best of our knowledge, this GWAS is one of the earliest studies that investigates important growth-related economic traits using the Illumina OvineSNP50 BeadChip in the Akkaraman sheep breed.

Growth related traits comprise an essential measuring basis of production performance in lambs, where these traits directly influence the economic benefit. It was reported that the heritability of body weight is moderate in sheep, with the heritability of birth weight and weaning weight ranging from 0.30 to 0.35 [25,26]. Importantly, genomic selection is preferred over conventional selection when favored traits have a low to moderate heritability. Thus, understating the complex molecular mechanisms and deciphering the important functional genes behind sheep liveweight are highly required.

Results from the current study revealed two genome-wide significant QTL regions on OAR2 that had not been previously reported as affecting liveweight and ADG in the Akkaraman sheep breed. It is important to note that the GWAS results demonstrated that rs*427117280* was involved in both T90 weight and ADG0_90. This result clearly indicates that the *rs427117280* might be a candidate SNP for liveweight and/or average daily gain in Akkaraman lamb. Supporting this finding, genotyping of SNP *rs427117280* in a further 373 Akkaraman lambs validated preliminary findings and showed association not only with ADG0_90 and T90, but also with ADG60_90 and ADG30_90. Association of other ADGs that were not found during the original GWAS may be due to the increased lamb number during the TaqMan genotyping study.

Alignment of the genetic and physical maps on the Oar_rambouillet_v1.0 genome assembly (Sheep QTL database, QTLdb) enabled comparison of the QTLs detected in our study with previously described QTL regions. Although, no QTLs have been detected on OAR2 for liveweight traits thus far, the significant SNP identified in this study did overlap with previously reported QTLs that related to the other traits in sheep [11]. Specifically, Campbell et al. [27] detected a significant QTL between 221.4–248.9 Mbp associated with bone density in Coopworth sheep which overlaps with our findings in the Akkaraman sheep. The fact that liveweight is so closely related to the growth of muscle, fat, and bone tissues supports the notion that SNP *rs427117280* is correctly associated with liveweight phenotypes. Further reinforcement of this concept is provided by a study completed on an Awassi and Merino cross, which suggested that a QTL for body weight at slaughter was present at 253 cM [28].

Assessment for nearby candidate genes associated with SNP *rs427117280* on OAR2 revealed the natriuretic peptide precursor-C (*NPPC*) gene. C-type natriuretic peptide (CNP) encoded by natriuretic peptide precursor-C (NPPC) has been shown to be important in skeletal development in mammals [29]. CNP is a member of the natriuretic peptide family, which binds to the cell-surface receptor Natriuretic Peptide Receptor 2 (NPR2). Interaction between CNP and NPR2 causes downstream synthesis of cyclic guanosine-3′,5′-monophosphate (cGMP) and regulates growth through osteogenesis and chondrogenesis [30]. Additionally, inhibition or overstimulation of genes encoding CNP result in severe dwarfism or skeletal overgrowth respectively [30]. Various researchers have previously suggested that sheep and cattle should be selected for a locus which contains the natriuretic peptide receptor-B (*NPR2*) gene, promoting CNPs involvement in animal development [31–36]. Lastly, mutations in either *NPR2*, *NPPC*, or nearby SNPs have been found to associate with different growth traits in humans [30], sheep [37] and cattle [38].

Another chromosome-wide suggestive QTN was detected for birth weight in Akkaraman lambs. The *rs406101124* intronic SNP in the Centrosomal Protein 152 (*CEP152*) gene was detected on OAR7 at the position of 63200764 G>A. Although no QTL was detected in the vicinity of *rs406101124* on the Sheep QTLdb, the *CEP152* gene has been found to associate with Seckel syndrome (OMIM 210600), characterized by severe intrauterine growth

retardation in humans [39,40]. It is well documented that birth weight is affected by intrauterine growth conditions, such as fetal genotype, in all mammalian livestock species [41]. Therefore, we suggest that SNP *rs406101124* in *CEP152* should be evaluated as a candidate SNP for selecting Akkaraman lambs with high birth weight. The intergenic *rs428733788* SNP that was suggested to be a chromosome-wide QTN for T30, is in close proximity to the A Disintegrin and Metalloproteinase thrombospondin type 1 motif 1 (*ADAMTS1*) gene. The region where *ADAMTS1* is located on OAR1 is enriched with previously detected QTLs for bone density in Coopworth sheep [27] and body weight at 20 weeks in the Charollais breed [42]. The ADAM and closely related ADAMTS1, is a family of proteolytic enzymes that possess sheddase function, which regulates shedding of membrane-bound proteins, growth factors, cytokines, ligands, and receptors [43]. ADAMTS1 modulates TGF–β (Transforming growth factor) and BMPs (Bone morphogenetic proteins), are responsible for growth of various tissues. Furthermore, ADAM based proteins can act as a NOTCH1 inhibitor, which is known to affect muscle mass and regenerative capacity [44]. It has been shown that the ADAMTS1 overexpression in macrophages is able to increase number of active satellite cells and enhance the regeneration of muscles in young animals [45]. Finally, the *rs430444267* SNP was suggested as a chromosome-wide QTN for ADG0_30, where this SNP lies within the MAM Domain Containing 2 (*MAMDC2*) gene. No literature was found to support association of *MAMDC2* with growth or skeletal muscle mass in animals. However, McRae et al. [41], detected two QTL in the Charollais sheep breed for backfat starting at the third lumbar vertebra that overlaps with our results.

## Conclusions

A strong association between a variant of SNP *rs427117280* and pre-weaning growth was demonstrated. Additionally, a set of candidate genes neighboring to the GWA signals were proposed here due to their potential influence on the investigated traits in Akkaraman lambs. Genomic selection may help to overcome the difficulties in selection of indigenous breeds that requires classical breeding programs where pedigree recording is a pre-requisite. The current results could be used as a foundation to guide and uncover biological mechanisms to develop markers underlying liveweight variation in multi-breed validation studies.

## Supporting information

**S1 Fig. MDS plot of Akkaraman lamb population.**
(JPG)

**S2 Fig. Quantile-quantile (Q-Q) plot of GWAS based on the random regression model.**
The x-axis and y-axis represent −log$_{10}$ transformed expected *p*-values and observed *p*-values, respectively. The dots indicate −log$_{10}$(*p*) of the SNPs and the diagonal line represents the expected values under the null hypothesis for no association.
(JPG)

**S1 Table. Measured liveweights and average daily gain values for Akkaraman lambs (in kg).**
(DOCX)

## Acknowledgments

Authors indebted to Ms. Codie Durfee (USA) and Dr. Mehmet Kizilaslan (Turkiye) for their helps during *in silico* analysis. The authors also thank the Betül Ziya Eren Genome and Stem Cell Centre (GENKÖK) at Erciyes University for providing laboratory facilities.

## Author Contributions

**Conceptualization:** Mehmet Ulas Cinar, Bilal Akyüz.

**Data curation:** Mehmet Ulas Cinar, Davut Bayram, Bilal Akyüz.

**Funding acquisition:** Bilal Akyüz.

**Methodology:** Korhan Arslan, Md Mahmodul Hasan Sohel, Fadime Daldaban, Esma Gamze Aksel.

**Writing – original draft:** Mehmet Ulas Cinar, Bilal Akyüz.

**Writing – review & editing:** Lindsay M. W. Piel, Stephen N. White.

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
