## [Decision Letter · Decision Letter 0]

15 Mar 2023

PONE-D-22-26401Genome-wide association study of early liveweight traits in fat-tailed Akkaraman lambsPLOS ONE

Dear Dr. Çınar,

Thank you for submitting your manuscript to PLOS ONE. After careful consideration, we feel that it has merit but does not fully meet PLOS ONE’s publication criteria as it currently stands. Therefore, we invite you to submit a revised version of the manuscript that addresses the points raised during the review process.

We look forward to receiving your revised manuscript.

Kind regards,

Muhammad Abdul Rehman Rashid, PhD

Academic Editor

PLOS ONE

Journal Requirements:

Reviewers' comments:

Reviewer's Responses to Questions

**Comments to the Author**

1. Is the manuscript technically sound, and do the data support the conclusions?

Reviewer #1: Yes

Reviewer #2: No

Reviewer #3: Partly

2. Has the statistical analysis been performed appropriately and rigorously? 

Reviewer #1: Yes

Reviewer #2: Yes

Reviewer #3: No

3. Have the authors made all data underlying the findings in their manuscript fully available?

Reviewer #1: Yes

Reviewer #2: Yes

Reviewer #3: No

4. Is the manuscript presented in an intelligible fashion and written in standard English?

Reviewer #1: Yes

Reviewer #2: No

Reviewer #3: Yes

5. Review Comments to the Author

Reviewer #1: Comments to the authors

PONE-D-22-26401 Genome-wide association study of early liveweight traits in fat-tailed Akkaraman lambs

Çinar et al.

In general, the manuscript is well written. The study was intended to identify SNPs associated with early live weight traits in fat-tailed Akkaraman lambs using GWAS. Line-specific comments are provided below:

Line 188: Is Animals and Phenotypic Data a subtitle?

Line 212-214: Sentence is repeated.

Line 227-228 and elsewhere: Suggest describing the parameters such as T90, and ADG0_90 on first use.

Line 234-237: The trait associated with SNP rs406101124 is birth weight and not T30, as per table 1. The candidate gene for this was not reported in table 1 either.

Line 237-240: The trait is not reported in the text.

Reviewer #2: Dear Authors,

Unfortunately, I cannot recommend your article for publication in PLOS ONE. This journal is designed to present research at the cutting edge of science. Your research certainly deserves attention. However, there are a number of remarks that do not allow us to consider it advanced. The number of animals (146) is too small for a full GWAS. The DNA chip you used is currently used primarily for routine genotyping. The search for associations at the modern level is performed on a consortium 600K chip or using whole genome sequencing data. There are too many mistakes and repetitions in the text, I started to make comments, but it turned out that there were too many of them. Here are just a few of the research results.

L215 – Is it Bonferroni correction? And “1.565 × 10−6 (0.05/31,936)” is genome-wide. Fix it.

L218 – In Table showing 5 SNP.

L228 – Don’t repeat data from table in text. Just analyze it.

L238 – Gene was localizing in intergenic region? Fix it.

L242 – Describe region of gene, if SNP localizing in it.

L250 – p-value not correcting by Bonferroni correction. It is threshold of p-value with significance level.

L253 – Why you are using 500 kbp distance?

L256 – What is the lines in graph?

L264 – You are fully repeat table data. In this case remove table.

L275 – “significantly heavier” wrong sentence. Change it.

L277 – Not. In statistic no tendencies. Significant difference presents or not.

In conclusion, I would recommend that you finalize the article and submit it for publication in a journal with a lower impact factor, where it can be published.

Regards,

Reviewer #3: OVERALL:

-The authors aimed at identifying SNPs associated with body weight in fat tail Akkaraman lambs. However, the number of sheep utilized for this analysis is low and could lead to false associations. Additionally, a low number of animals reduces the chances of identifying regions controlling the traits of interest. This could be one of the reasons why the authors only found one SNP on chr 2 associated with weaning weight at day 90 and average daily gain (ADG) from day 0 to day 90.

INTRODUCTION:

-Line 33: Check for typos (grammatical errors).

MATERIALS AND METHODS:

-Please include the protocol number of the approved animal use protocol (i.e., IACUC number) utilized in this study.

-The authors mentioned that pedigree was unknown. However, you can estimate this from SNP data. Please include the G matrix in the association model.

- Please include a better description of the model used for the association analysis. Please use a mixed model and explain the components of the model (fixed effects, random effects, etc). Please also include sex if male and female lambs were evaluated and re-analyze the data.

- It is possible that authors won’t find the same results after adjusting the association model. Please revise the data and results.

- Submit your SNP data to ENSEMBL/NCBI and provide number. This information must be part of the manuscript (Genotyping section).

- Revise abstract, results, discussion, and conclusion sections after adjusting the association model.

6. PLOS authors have the option to publish the peer review history of their article (what does this mean?). If published, this will include your full peer review and any attached files.

Reviewer #1: No

Reviewer #2: No

Reviewer #3: No

---

## [Author Response · Author response to Decision Letter 0]

10 Aug 2023

Reviewer #1: Comments to the authors

PONE-D-22-26401 Genome-wide association study of early liveweight traits in fat-tailed Akkaraman lambs Çinar et al.

Comment: In general, the manuscript is well written. The study was intended to identify SNPs associated with early live weight traits in fat-tailed Akkaraman lambs using GWAS. Line-specific comments are provided below:

Response: Thank you for comment.

Comment: Line 188: Is Animals and Phenotypic Data a subtitle?

Response: The sentence was corrected.

Comment: Line 212-214: Sentence is repeated.

Response: Additional sentence was deleted.

Comment: Line 227-228 and elsewhere: Suggest describing the parameters such as T90, and ADG0_90 on first use.

Response: The sentence was added in the section of “Animals and phenotypic data”. Additionally, footnote #2 is covering the explanation of ADGs. 

Comment: Line 234-237: The trait associated with SNP rs406101124 is birth weight and not T30, as per table 1. The candidate gene for this was not reported in table 1 either.

Response: The phenotype “birth weight” was added in the revised manuscript. Since SNP rs406101124 was located within the CEP152 gene. The candidate gene FBN1 was given in the text and table in the revised version. 

Comment: Line 237-240: The trait is not reported in the text.

Response: The trait information was added in the revised version. 

Reviewer #2: Dear Authors,

Comment: Unfortunately, I cannot recommend your article for publication in PLOS ONE. This journal is designed to present research at the cutting edge of science. Your research certainly deserves attention. However, there are a number of remarks that do not allow us to consider it advanced. The number of animals (146) is too small for a full GWAS. The DNA chip you used is currently used primarily for routine genotyping. The search for associations at the modern level is performed on a consortium 600K chip or using whole genome sequencing data. There are too many mistakes and repetitions in the text, I started to make comments, but it turned out that there were too many of them. Here are just a few of the research results.

Response: We agree your comment. But Akkaraman is a native breed that recording of matings could be sometimes biased. We have started this reseach based on farmer’s data but we also understood that inbreeding was relatively higher in the population. Additionally, lack of breeders organization for specific native breeds, rams generally has been used repetetively for mating in years in Turkish sheep husbandry system. Thus, relatidness of half-sib could be analyzed only with whole genotyping. We have started with 192 lambs. However, due to inbreeding 46 lambs were excluded from the analysis. 

Furthermore, our results based on 146 animals was validated uding TaqMan genotyping assay in 400 animal population. Results showed that the rs427117280 SNP was found to be associated with weaning weight at day 90 and average daily gain (ADG) from day 0 to day 90. Our experiments is ongoing that covers other native breeds of Turkiye for validation of rs427117280 SNP with early growth traits.

600K chip could be a good option, however we followed similar researchers and literature during the experimental design. Following articles are showing that similar number of animals using 50K chip published for ovine GWA studies. In further studies, we are planning to enlarge animal population and to utilize 600K ovine chip. 

 Hernández-Montiel, W., Martínez-Núñez, M. A., Ramón-Ugalde, J. P., Román-Ponce, S. I., Calderón-Chagoya, R., & Zamora-Bustillos, R. (2020). Genome-wide association study reveals candidate genes for litter size traits in pelibuey sheep. Animals, 10(3), 434.

 James, C., Pemberton, J. M., Navarro, P., & Knott, S. (2022). The impact of SNP density on quantitative genetic analyses of body size traits in a wild population of Soay sheep. Ecology and Evolution, 12(12), e9639.

 Almasi, M., Zamani, P., Mirhoseini, S. Z., & Moradi, M. H. (2021). Genome-wide association study for postweaning weight traits in Lori-Bakhtiari sheep. Tropical Animal Health and Production, 53, 1-8.

 Hernández-Montiel, W., Martínez-Núñez, M. A., Ramón-Ugalde, J. P., Román-Ponce, S. I., Calderón-Chagoya, R., & Zamora-Bustillos, R. (2020). Genome-wide association study reveals candidate genes for litter size traits in pelibuey sheep. Animals, 10(3), 434.

 Mastrangelo, S., Ben Jemaa, S., Sottile, G., Casu, S., Portolano, B., Ciani, E., & Pilla, F. (2019). Combined approaches to identify genomic regions involved in phenotypic differentiation between low divergent breeds: Application in Sardinian sheep populations. Journal of Animal Breeding and Genetics, 136(6), 526-534.

 Hernández-Montiel, W., Cob-Calan, N. N., Cahuich-Tzuc, L. E., Rueda, J. A., Quiroz-Valiente, J., Meza-Villalvazo, V., & Zamora-Bustillos, R. (2022). Runs of Homozygosity and Gene Identification in Pelibuey Sheep Using Genomic Data. Diversity, 14(7), 522.

Comment: L215 – Is it Bonferroni correction? And “1.565 × 10−6 (0.05/31,936)” is genome-wide. Fix it.

Response: The sentence was fixed. Yes, it is Bonferroni corrected values, mentioned in Genome-Wide Association Study sub-section.

Comment: L218 – In Table showing 5 SNP

Response: The sentence was corrected in the revised manuscript. 

Comment: L228 – Don’t repeat data from table in text. Just analyze it.

Response: Sentences were removed and table 1 was referred.

Comment: L238 – Gene was localizing in intergenic region? Fix it.

Response: The rs428733788 SNP was found to be intergenic, however, ADAMTS1 was identified as closest candidate gene to be responsible for related growth trait. The sentence was rephrased in order to avoid misunderstanding.

Comment: L242 – Describe region of gene, if SNP localizing in it.

Response: The gene was given in the rephrased sentence. 

Comment: L250 – p-value not correcting by Bonferroni correction. It is threshold of p-value with significance level.

Response: Thank for comment. Footnote section was corrected and Bonferroni term was removed. 

Comment: L253 – Why you are using 500 kbp distance?

Response: According to literature, we have observed that different distance were used between detected SNP and candidate gene among various experiments. Here we listed some literature that were identified and discussed candidate genes in different distances to identified SNP:

 Esmaeili-Fard SM, Gholizadeh M, Hafezian SH, Abdollahi-Arpanahi R (2021) Genome-wide association study and pathway analysis identify NTRK2 as a novel candidate gene for litter size in sheep. PLOS ONE 16(1): e0244408. (1 Mb distance was investigated)

 Bolormaa, S., Swan, A.A., Stothard, P. et al. A conditional multi-trait sequence GWAS discovers pleiotropic candidate genes and variants for sheep wool, skin wrinkle and breech cover traits. Genet Sel Evol 53, 58 (2021). (0.97 Mb distance was investigated)

 Wang Z, Zhang H, Yang H, Wang S, Rong E, Pei W, et al. (2014) Genome-Wide Association Study for Wool Production Traits in a Chinese Merino Sheep Population. PLoS ONE 9(9): e107101. (0.40 Mb distance was investigated)

 Sutera AM, Moscarelli A, Mastrangelo S, Sardina MT, Di Gerlando R, Portolano B, Tolone M. Genome-Wide Association Study Identifies New Candidate Markers for Somatic Cells Score in a Local Dairy Sheep. Front Genet. 2021 Mar 22;12:643531. (0.50 Mb distance was investigated)

As we can see there some more literature that used various range surrounding each significant SNP. There is no specific range has been reported in literature for ovine GWAS studies. 

Comment: L256 – What is the lines in graph?

Response: Blue line represents chromosome-wide significance while red one represents genome-wide significance levels.

Comment: L264 – You are fully repeat table data. In this case remove table.

Response: The table 2 was modified as supplementary table give as Table S1.

Comment: L275 – “significantly heavier” wrong sentence. Change it.

Response: The sentence was modified and highlighted as yellow. 

Comment: L277 – Not. In statistic no tendencies. Significant difference presents or not.

Response: Thanks for comment: The sentence was modified: 

From: Notably, statistical analysis exhibited a trend toward higher significance values between genotypes and measured traits in the larger Akkaraman sheep population (p<0.1).

To: Notably, statistical analysis exhibited a significant association between genotypes and measured traits in the larger Akkaraman sheep population (p<0.1).

Comment: In conclusion, I would recommend that you finalize the article and submit it for publication in a journal with a lower impact factor, where it can be published. 

Response: We are aware of animal number is relatively small. However, though Akkraman is consisting of approximately the 45% of whole sheep population in Turkiye, it is accepted as extensively bred local sheep breed. Therefore, lack of herd book and breed organization is limiting to access regular phenotyping data. Additionally, GWAS comprising similar or lower number of animals have been publishing in PloS One and other scientific journals. 

 Esmaeili-Fard SM, Gholizadeh M, Hafezian SH, Abdollahi-Arpanahi R (2021) Genome-wide association study and pathway analysis identify NTRK2 as a novel candidate gene for litter size in sheep. PLOS ONE 16(1): e0244408. (n=91)

 Kominakis A, Tarsani E, Hager-Theodorides AL, Mastranestasis I, Gkelia D, Hadjigeorgiou I (2021) Genetic differentiation of mainland-island sheep of Greece: Implications for identifying candidate genes for long-term local adaptation. PLoS ONE 16(9): e0257461. (n=237)

 Tuersuntuoheti M, Zhang J, Zhou W, Zhang C-l, Liu C, Chang Q, et al. (2023) Exploring the growth trait molecular markers in two sheep breeds based on Genome-wide association analysis. PLoS ONE 18(3): e0283383. (n=184)

Regards,

Reviewer #3: OVERALL:

-The authors aimed at identifying SNPs associated with body weight in fat tail Akkaraman lambs. However, the number of sheep utilized for this analysis is low and could lead to false associations. Additionally, a low number of animals reduces the chances of identifying regions controlling the traits of interest. This could be one of the reasons why the authors only found one SNP on chr 2 associated with weaning weight at day 90 and average daily gain (ADG) from day 0 to day 90.

INTRODUCTION:

Comment: Line 33: Check for typos (grammatical errors).

Response: The sentence was corrected.

MATERIALS AND METHODS:

Comment: Please include the protocol number of the approved animal use protocol (i.e., IACUC number) utilized in this study.

Response: The following sentence was added into material and methods part. “All experimental procedures were performed in accordance with the guidelines of the Local Ethics Committee for Animal Experiments at Erciyes University (#13/130-13.11.2013)”

Comment a: The authors mentioned that pedigree was unknown. However, you can estimate this from SNP data. Please include the G matrix in the association model.

Comment b: Please include a better description of the model used for the association analysis. Please use a mixed model and explain the components of the model (fixed effects, random effects, etc). Please also include sex if male and female lambs were evaluated and re-analyze the data.

Response a and b: PCA based association analysis is the most popular approach for GWAS due to its simplicity and effectivenes (Zhang and Pan, 2015). Linear mixed model based associations are also used for GWAS due to their capacity of handling population stratification and criptic relatedness (Chen and Abecasis, 2007). However, due to several reasons such as their treatment to the estimations for fixed versus random effects, simplicity and effectivenes PCA-based association is defined as the ‘gold standard’ (Elhaik, 2022). To account for the mild population stratification, we used a PCA based association test, where top 10 PCs were fitted in the model for the GWAS of each trait. The model of PCA-based linear regression for GWAS is provided below: 

y= µ+Xβx+Zβz+ e

Where y is the vector of observations, µ is the intercept, X and Z are the design matrices mapping SNP effect and 10 PCs constructed by PCA from a large number of genetic variants to the observations, respectively (Lee et al., 2009). βz is the effect of eigenvectors and βx is the effect of SNP being tested for association. e ~ MVN (0, σ2I) is the error term. The goal of an association analysis is to test the null hypothesis H0: γ1 = 0.

 Zhang Y, Pan W. Principal component regression and linear mixed model in association analysis of structured samples: competitors or complements? Genet Epidemiol. 2015 Mar;39(3):149-55. doi: 10.1002/gepi.21879. Epub 2014 Dec 23. PMID: 25536929; PMCID: PMC4366301. 

 Elhaik, E. Principal Component Analyses (PCA)-based findings in population genetic studies are highly biased and must be reevaluated. Sci Rep 12, 14683 (2022). https://doi.org/10.1038/s41598-022-14395-4

 Chen, W.-M., Abecasis, G.R., 2007. Family-based association tests for genome-wide association scans. Am. J. Hum. Genet. 81, 913–926.

Comment: It is possible that authors won’t find the same results after adjusting the association model. Please revise the data and results.

Response: Due to explanation which is found above we have not revised the data.

Comment: Submit your SNP data to ENSEMBL/NCBI and provide number. This information must be part of the manuscript (Genotyping section).

Response: SNPs those were given as results are already abundant in the sheep reference genome. Their rs numbers are unique and represents each SNP.

Comment: Revise abstract, results, discussion, and conclusion sections after adjusting the association model.

Response: Due to explanation which is found above we have not revised the data.

---

## [Editor Report · Decision Letter 1]

6 Sep 2023

Genome-wide association study of early liveweight traits in fat-tailed Akkaraman lambs

PONE-D-22-26401R1

Dear Dr. Çınar,

We’re pleased to inform you that your manuscript has been judged scientifically suitable for publication and will be formally accepted for publication once it meets all outstanding technical requirements.

Kind regards,

Muhammad Abdul Rehman Rashid, PhD

Academic Editor

PLOS ONE

---

## [Editor Report · Acceptance letter]

15 Sep 2023

PONE-D-22-26401R1 

Genome-wide association study of early liveweight traits in fat-tailed Akkaraman lambs 

Dear Dr. Cinar:

I'm pleased to inform you that your manuscript has been deemed suitable for publication in PLOS ONE. Congratulations! Your manuscript is now with our production department. 

Kind regards, 

on behalf of

Dr. Muhammad Abdul Rehman Rashid 

Academic Editor

PLOS ONE